# Molecular Phylogenies of Leeches and Haemoparasites Infecting Freshwater Turtles in Aquatic Ecosystems of Northern Africa Suggest Phylogenetic Congruence between *Placobdella costata* Sensu Lato and *Haemogregarina stepanowi* Sensu Lato

**DOI:** 10.3390/microorganisms11061584

**Published:** 2023-06-15

**Authors:** Olivier Verneau, Sirine Melliti, Latifa Kimdil, El Hassan El Mouden, Mohamed Sghaier Achouri, Rachid Rouag

**Affiliations:** 1Centre de Formation et de Recherche sur les Environnements Méditerranéens, University of Perpignan Via Domitia, UMR 5110, F-66860 Perpignan, France; sirine.melliti@univ-perp.fr (S.M.); latifa.kimdil@univ-perp.fr (L.K.); 2CNRS, Centre de Formation et de Recherche sur les Environnements Méditerranéens, UMR 5110, F-66860 Perpignan, France; 3Unit for Environmental Sciences and Management, North-West University, Potchefstroom Campus, Private Bag X6001, Potchefstroom 20520, South Africa; 4Laboratoire de Diversité, Gestion et Conservation des Systèmes Biologiques, LR18ES06, Faculté des Sciences de Tunis, Université Tunis-El Manar, Tunis 2092, Tunisia; mohamed.achouri@gmail.com; 5Laboratory of Water, Biodiversity and Climatic Change, Faculty of Sciences, Semlalia, Cadi Ayyad University, Marrakech 40000, Morocco; elmouden@uca.ac.ma; 6Laboratoire de Recherche Biodiversité et Pollution des Ecosystèmes, Faculté des Sciences de la Nature et de la Vie, Université Chadli Bendjedid El-Tarf, BP73, El-Tarf 36000, Algeria; rachid_rouag@yahoo.fr

**Keywords:** Testudines, Mediterranean pond turtle, European pond turtle, COI, Apicomplexa, haemogregarines

## Abstract

Haemogregarines are blood parasites with a life-cycle involving a vertebrate as the intermediate host and an invertebrate as the definitive host and vector. Extensive phylogenetic investigations based on 18S-rRNA gene sequences have shown that *Haemogregarina stepanowi* (Apicomplexa: Haemogregarinidae) is able to infest a large diversity of freshwater turtle species, including the European pond turtle *Emys orbicularis*, the Sicilian pond turtle *Emys trinacris*, the Caspian turtle *Mauremys caspica*, the Mediterranean pond turtle *Mauremys leprosa*, and the Western Caspian turtle *Mauremys rivulata*, among others. From the same molecular markers, *H. stepanowi* is further considered to be a complex of cryptic species predisposed to infect the same host species. While *Placobdella costata* is known to be the unique vector of *H. stepanowi*, it is only recently that independent lineages within *P. costata* have been illustrated—suggesting the presence of at least five unique leech species across Western Europe. The aims of our study were therefore to investigate from mitochondrial markers (COI) the genetic diversity within haemogregarines and leeches infecting freshwater turtles of the Maghreb, in order to identify processes of parasite speciation. We showed that *H. stepanowi* consists of at least five cryptic species in the Maghreb, while two *Placobella* species were identified in the same area. Although an Eastern–Western speciation pattern was apparent for both leeches and haemogregarines, we cannot make definitive conclusions regarding co-speciation patterns between parasites and vectors. However, we cannot reject the hypothesis of a very strict host–parasite specificity within leeches.

## 1. Introduction

Haemogregarines are blood parasites of the phylum Apicomplexa with a life-cycle involving a vertebrate as an intermediate host and an invertebrate as a definitive host and vector. They are currently classified into four families: the Haemogregarinidae which comprise three genera, namely *Haemogregarina*, *Cyrilia*, and *Desseria*; the Hepatozoidae with a single genus, *Hepatozoon*; the Karyolysidae with two genera, namely *Hemolivia* and *Karyolysus*; and the Dactylosomatidae, also with two genera, *Babesiosoma* and *Dactylosoma* [1]. Whereas the intermediate host encompasses all types of vertebrates, i.e., fishes, mammals, amphibians, reptiles, and birds, the vector host encompasses solely leeches and acarines. Since Siddall [2] and his work based on a cladistic approach of morphological characters, the systematics of haemogregarines have been greatly improved. Siddal [2] was the first to show *Haemogregarina* as a polyphyletic taxon. He thus transferred all species infecting snakes, lizards, crocodilians, and birds, which were originally described as *Haemogregarina* spp., into the genus *Hepatozoon* and retained all other species infecting fishes, amphibians, and chelonians in the genus *Haemogregarina* sensu lato. In addition, based on other informative characters, the distinction of *Haemogregarina* sensu stricto was allowed to accommodate mostly *Haemogregarina* spp. infecting chelonians. Smith [3] completed the classification revised by Siddall [2] in transferring certain haemogregarine species from one genus to another. Following the advent of molecular data in phylogenetics by the end of the 1990s, certain haemogregarines have been investigated for their phylogenetic position within the Apicomplexa—more particularly, species of *Hepatozoon* [4,5,6]. The first comprehensive study that included the taxa of the four recognized families was conducted by Barta et al. [7], although *Hepatozoon* species were still over-represented in their phylogeny in comparison to the other taxa, namely *Hemolivia*, *Dactylosoma*, *Bebesiosoma*, and *Haemogregarina*. Later on, the non-monophyly of *Hepatozoon* species was evidenced [8,9,10] following an investigation of the phylogenetic relationships of species of the four haemogregarine familes. In addition to the paraphyly of *Hepatozoon* species, the paraphyly of the Karyolysidae was also illustrated [11].

While the monophyly of particular genera and their relationships within the Adeleorina remain to be investigated using molecular markers, numerous questions have been raised regarding the genetic diversity of haemogregarines infecting vertebrate hosts, especially freshwater chelonians. Extensive phylogenetic investigations during the ten past years on *Haemogregarina stepanowi* Danilewsky 1885 have demonstrated that this species is able to infest a large diversity of the turtle species of North Africa and Western Europe: among others, the European pond turtle *Emys orbicularis* (Linnaeus); and the Sicilian pond turtle *Emys trinacris* Fritz, Fattizzo, Guicking, Tripepi, Pennisi, Lenk, Joger, and Wink of the Emydidae; the Caspian turtle *Mauremys caspica* (Gmelin); the Mediterranean pond turtle *Mauremys leprosa* (Schweigger); and the Western Caspian turtle *Mauremys rivulata* (Valenciennes) of the Geoemydidae [12,13,14]. Although the intermediate host specificity for this parasite was believed to be very low, Attia El Hili et al. [15] further evidenced from 18S-rRNA gene sequences the occurrence of distinct species of *Haemogregarina* within the same chelonian host species, in both *M. leprosa* and *E. orbicularis* turtles, thus suggesting coinfection. Based on genetic data, one of these haemogregarine species could be considered with certainty to be *H. stepanowi*. All of these results were soon confirmed from an analysis of target COI sequences obtained from the same isolates of blood samples [16]. Nine distinct COI haplotypes were revealed corresponding to three molecular lineages, each corresponding to a distinct *Haemogregarina* species, as previously advocated from an analysis of nuclear markers [15]. A similarly high *Haemogregarina* genetic diversity has since been evidenced from mitochondrial markers within the Savanna side-necked turtle *Podocnemis vogli* Müller [17], as well as from nuclear markers within the yellow-spotted Amazon river turtle *Podocnemis unifilis* Troschel [18] and *E. orbicularis* [19]. Due to the low host specificity of *H. stepanowi* against its intermediate host on the one hand [12,13,14] and the high diversity of *Haemogregarina* spp. in certain turtle species on the other [16,17,18,19], the host specificity of *Haemogregarina* could act against leeches—namely *Placobdella* as the definitive host—as claimed earlier by Attia El Hili et al. [16].

*Placobdella* is a widespread genus of blood-feeding leeches belonging to the family Glossiphoniidae (Clitellata, Hirudinea, Rhynchobdellida). It includes 25 recognized species [20], most of which display a Nearctic distribution [21]. *Placobdella* leeches are common ectoparasites of freshwater vertebrates. It has been shown that host specificity differs among *Placobdella* spp. Certain leeches exhibit an opportunistic generalist behavior with a wide range of host species: for instance, *Placobdella montifera* Moore 1906 [22,23] and *Placobdella phalera* (Graf 1899) [24] which infest a great many species; *Placobdella multilineata* Moore 1953 which has been reported in 17 species and subspecies of reptiles, such as alligators, crocodiles, snakes, and turtles, as well as in amphiumas [25]; and *Placobdella ornata* (Verrill 1872) which infests turtles, as well as fish, amphibians, birds, and humans [22]. In contrast, certain other leeches exhibit a very strict host specificity. This is the case for *Placobdella cryptobranchii* (Johnson and Klemm 1977) [26] which has been reported from a single amphibian host species, i.e., the Ozark hellbender *Cryptobranchus alleganiensis bishopi* Grobman [22,24,26], although Moser et al. [27] claimed an additional host species, i.e., the Mudpuppy *Necturus maculosus* (Rafinesque). This was also the case for *Placobdella appalachiensis* Moser and Hopkins 2014 which was reported from a single host species, i.e., the Eastern hellbender *Cryptobranchus alleganiensis* (Daudin) [28]. *Placobdella costata* (Fr Müller 1846), the type species of the genus *Placobdella*, was considered as the only European representative of the genus until Soors et al. [29] reported *Placobdella ornata* (Verril 1872) in Europe following the introduction of its host. While it was first thought to infect a single turtle species, i.e., *E. orbicularis* across its entire geographical distribution, from Western Europe to Russia [30], as well as in Algeria [31] and Tunisia [15], *P. costata* is now known to infect certain other turtles. This has indeed been reported from *M. caspica* in Iran [32,33], from *E. trinacris* in Sicilia [34], and from *M. leprosa* in Spain [35,36], Morocco [37], and Tunisia [15]. While it is now well received from the investigation of molecular markers that *Placobdella* is a monophyletic genus, certain cryptic diversity was evidenced within several species, among which are *Placobdella mexicana* Moore 1898 and *Placobdella ringueleti* López-Jiménez and Oceguera-Figueroa 2009 [38]. Based on COI and ITS1 genetic variations, Kvist et al. [39] also illustrated seven independent lineages within *P. costata*, suggesting at least five unique leech species across Central and Eastern Europe, as well as in Algeria. Independent phylogenetic relationships were shown among Italian populations of *P. costata* and their two host species *E. orbicularis* and *E. trinacris* [40]; thus, Kvist et al. [39] concluded that species diversity and speciation within *P. costata* were likely not the result of speciation, isolation, or dispersal of the host species.

Hence, considering the low specificity of *H. stepanowi* towards its intermediate chelonian hosts, the high diversity of *Haemogregarina* spp. within turtles, as well as the cryptic species diversity within *P. costata*, we aimed to investigate the genetic diversity within haemogregarines and leeches infecting North African freshwater turtles. Our objectives were to compare phylogenetic patterns between hosts and their parasites in order to identify processes of parasite speciation.

## 2. Materials and Methods

Field-work investigations were carried out on 2019 and 2020, mainly in freshwater aquatic environments of Morocco, Algeria, and Tunisia suitable for *E. orbicularis* and *M. leprosa* (see Figure 1 and Table 1 for GPS localities). Sampling sessions were conducted during the peak activity period of both turtles, from March to November. Traps were baited with fish and set in waters with plastic bottles to maintain them at the surface water and to prevent turtles’ drowning. They were firmly attached to the bank of the water bodies with a solid rope and checked every day, usually between one to three days, before being removed. Trapped turtles were marked individually on the marginal scutes for further capture–mark–recapture procedures. They were sexed, measured, weighted, and checked for ectoparasites. Leeches usually occur at the base of the anterior and/or posterior limbs. When present, they were collected and preserved in 70% alcohol for molecular systematic studies. Blood was also collected from each specimen from the dorsal coccygeal vein running in the midline of the turtle’s tail with the help of 1-mL insulin syringes. A drop of blood was immediately spread out on a glass slide, fixed with a few drops of methanol after air drying, and stored temporarily until staining in the lab. Of the remaining blood, one half was stored in 90% ethanol for molecular systematic studies of the host and their haemoparasites, and the second half was preserved into two distinct Microtainer^®^ tubes with anticoagulant that were immediately frozen in liquid nitrogen before being transported to the lab where they were finally stored at −80 °C for further biochemical analyses. Investigated turtles were released at the same place of capture in the field.

Blood smears on slides were stained with 10% Giemsa for 20 min in the laboratory, air dried, and examined by optical microscopy using a ×40 objective for the first screening. A Leica digital camera using a ×100 objective was used to capture images for biometric measurements and developmental-stage identification according to Dvoráková et al. [12,41]. A total of 422 blood smears were prepared and examined, including 406 from *M. leprosa* collected at 12 localities of Morocco, six of Algeria, and 14 of Tunisia, and 16 from *E. orbicularis* collected at five localities of Algeria.

DNA extractions were performed with the E.Z.N.A Tissue DNA Kit (Omega bio-tek, Norcross, GA, USA) following recommendations of the supplier for blood and tissue samples preserved in ethanol. DNA extracts were resuspended in approximately 200 µL of Tris-EDTA 1X elution buffer before use for PCR. The cytochrome c oxidase I (COI) of leeches was amplified using the forward LCO Plac 5′-AYTCAACTAATCAYAAAGAYATTGG-3′ and reverse HCO Plac 5′-TADACTTCWGGRTGACCAAAAAATCA-3′ primers which were designed for *Placobdella* spp. [38]. The COI of haemogregarines was amplified using the forward HemoFor4 5′-TGGACATTATACCCACCTTTAAG-3′ and reverse HemoRev4 5′-ATACAACCCATAGCTAGTATCAT-3′ primers which were designed for *Haemogregarina* spp. [16]. PCR assays of COI were conducted in a final volume of 25 μL comprising 1X buffer, 1.5 mM MgCl_2_, 0.2 mM dNTPs, 0.4 mM primers, 0.75 units GoTaq Polymerase (Promega, Charbonnières-les-Bains, France), and DNA (3 µL) under the following conditions: an initial step of 5 min at 95 °C for long denaturation; followed by 35 cycles of 30 s at 95 °C for denaturation, 30 s at 48 °C for annealing, and 1 min at 72 °C for elongation; with a final step of 10 min at 72 °C for terminal extension. The success of the PCR amplifications was controlled following electrophoresis in gels of 1% agarose stained with ethidium bromide. When amplifications were successful, they yielded PCR fragments of approximately 700 bp for the COI of leeches and 465 bp for the COI of haemogregarines which were subsequently sequenced by the Genoscreen Company (Lille, France).

COI chromatograms were edited with Chromas 2.2.6 (Technelysium Pty Ltd., Brisbane, Australia), and the resulting new sequences obtained for leeches and haemogregarines were respectively grouped in two distinct files with certain others that were extracted from GenBank. The final COI leech data set included 78 COI sequences corresponding to 23 distinct *Placobdella* species on the one hand and two *Helobdella* species used for outgroup comparisons on the other. At this stage, *P. costata* was considered as a single species and referred as *P. costata* sensu lato. Conversely, the final COI protozoan data set included 86 COI sequences corresponding to seven species of Piroplasmida, 17 species of Haemosporida, and 30 species of Eucoccidiorida, including nine putative *Haemogregarina* species from two South American turtles, and *Scrippsiella sweeneyae* Balech 1965 used for outgroup comparisons [16]. At this stage, *H. stepanowi* was considered as a single species and referred as *H. stepanowi* sensu lato. The COI sequences included in both data sets were aligned independently using Clustal W [42] implemented in MEGA7 [43] under default parameters.

For the Bayesian analysis, a GTR + I + G model was selected independently for the two COI partitions following the hierarchical likelihood ratio tests (hLRTs) implemented in Modeltest 3.06 [44]. Therefore, six types of substitutions and invariable-gamma rates with four gamma rate categories were applied for each partition, evolutionary parameters being estimated independently. The leech COI partition included 689 characters, while the protozoan COI partition included 448 characters. Bayesian analyses were run using MrBayes 3.04b [45], with four chains running for five million generations and sampled every 100 cycles. Consensus trees for both data sets were drawn, after removing the first 5000 trees as the burn-in phase, and opened with TreeGraph 2 [46]. These were converted into a phylogram, and only nodes supported by more than 0.75 posterior probabilities were shown.

For the parsimony approach, a bootstrap test with 1000 replicates was applied for each data set following a heuristic search under PAUP* Version 4.0 b10 [47] on the parsimony-informative characters, gaps being treated as missing data. The nearest neighbor interchange (NNI) branch-swapping option was used to find trees of minimal length. In order not to weight down the illustrated trees, only bootstrap proportions higher than 75% were indicated next to the branches within the clades *P. costata* sensu lato and *H. stepanowi* sensu lato.

For the distance analysis, a bootstrap test with 5000 replicates was applied for each data set. Minimum evolution (ME) trees were searched using the close-neighbor-interchange (CNI) algorithm [48] at a search level of one for each replication based on Kimura two-parameter distances [49] that were computed under the pairwise-deletion option. Evolutionary analyses were conducted in MEGA7 [43]. Regarding parsimony, only bootstrap proportions higher than 75% were indicated next to the branches for the two clades of interest.

A total of 22 COI sequences from *Placobdella rugosa* (Verrill 1874) were retrieved from GenBank in order to investigate the molecular threshold within *Placobdella*. All of these sequences correspond to distinct leech specimens that were sampled at separate sites across Canada, which represents most of the geographic range of that species; thus, we expected that the intra-average distances measured within all these specimens may reflect the average genetic variations within each *Placobdella* species. Pairwise p-distances were calculated within *P. rugosa* and the average distance with its standard deviation (SD) was then estimated and given with the minimum and maximum distances between all investigated specimens. Finally, pairwise p-distances were calculated within and between haploclades of *P. costata* sensu lato to explore the species diversity.

The interspecific average pairwise COI p-distance was estimated to about 2.1% between two closely related valid distinct species, namely *Eimeria lancasterensis* Joseph 1969 and *Eimeria sciurorum* Galli-Valerio 1922, and further considered as the molecular threshold within *Haemogregarina* [16]. Therefore, pairwise p-distances were calculated within and between haploclades of *H. stepanowi* sensu lato to explore the species diversity based on the molecular threshold.

## 3. Results

Following field-work investigations, *M. leprosa* was sampled in 12 localities of Morocco, among which five showed the presence of leeches attached to turtles. Following blood observations, haemogregarines were observed within turtles in five localities: only three of them had leeches, the other two did not. The prevalence of haemogregarine infection varied from approximately 11% to 80% in localities where turtles were found to be infected (Table 1). In Algeria, *M. leprosa* was sampled in six localities, among which three showed the presence of leeches attached to turtles. Following blood observations, haemogregarines were observed within turtles in five localities: only three of them had leeches, the other two did not. The prevalence of haemogregarine infection varied from approximately 25% to 75% in localities where turtles were found to be infected. *Emys orbicularis* was also sampled in five localities of Algeria—three times in syntopy with *M. leprosa*—among which three localities showed the presence of leeches attached to turtles. Haemogregarines were observed within turtles in three localities, a single one had leeches, the other two did not. The prevalence of haemogregarine infection varied from approximately 50% to 75% in localities where turtles were found to be infected (Table 1). In Tunisia, *M. leprosa* was sampled in 14 localities, among which three showed the presence of leeches attached to turtles. Following blood observations, haemogregarines were observed within turtles in three localities: only two of them had leeches, the last one did not. The prevalence of haemogregarine infection varied from approximately 58% to 67% in localities where turtles were found to be infected (Table 1).

In total, 31 leeches were processed for their COI, 11 collected from *M. leprosa* sampled from four localities of Morocco, 15 collected from *M. leprosa* and *E. orbicularis* sampled from six localities of Algeria, and five collected from *M. leprosa* sampled from three localities of Tunisia (Table 2). They were characterized by six distinct haplotypes, C8a to C8c in Morocco, C5a to C5c in Algeria, and C5a to C5b in Tunisia. According to the phylogenetic reconstruction shown in Figure 2, *P. costata* sensu lato constitutes a robust clade that can be split into six distinct lineages, namely haploclades C1 + C7, C3, C4, C5, and C6 based on [39], and Haploclade C8 based on the present results. Conversely, due to technical difficulties in amplifying the COI of haemogregarines from blood samples, only 13 sequences were obtained from *M. leprosa* sampled from two localities of Morocco, from one locality of Algeria, and from two localities of Tunisia (Table 3). These were characterized by nine distinct haplotypes: 4a to 4d in Morocco, 1c in Algeria, and 3g to 3i and 5a in Tunisia. According to the phylogenetic reconstruction shown in Figure 3, *H. stepanowi* sensu lato constitutes a robust clade that can be split into five distinct lineages, namely haploclades 1 to 3 based on Attia El Hili et al. [16], and haploclades 4 and 5 based on the present results.

Regarding the average p-distance that was estimated within *P. rugosa*, the COI molecular threshold can be considered as approximately 0.90% pairwise substitutions ± 0.0042 within *Placobdella* (Table 4). Therefore, according to the average p-distances that were estimated within haploclades of *P. costata* sensu lato on the one hand, which varied from 0.30% for the minimal average distance to 1.44% for the maximal average distance (Table 4), and between haploclades of *P. costata* sensu lato on the other, which varied from 4.19% for the minimal average distance to 8.15% for the maximal distance (Table 5), each haploclade can be considered as a distinct species, which is in agreement with Kvist et al. [39]. If Haploclade C6 can be considered as *P. costata* sensu stricto according to Kvist et al. [39] (see Figure 2), leeches that were collected from *M. leprosa* and *E. orbicularis* of Algeria and Tunisia can be regarded as the undescribed species corresponding to Haploclade C5 that was previously identified [39]. Similarly, leeches that were collected from *M. leprosa* of Morocco can be regarded as another undescribed species corresponding to Haploclade C8 identified here for the first time.

Considering the importance of molecular COI of 2.1% pairwise substitutions [16], the average p-distances that were estimated within haploclades of *H. stepanowi* sensu lato on the one hand, which varied from 0.43% for the minimal average distance to 1.61% for the maximal average distance (Table 6), and between haploclades of *H. stepanowi* sensu lato on the other, which varied from 6.71% for the minimal average distance to 8.21% for the maximal distance (Table 7), suggest that each haploclade can be considered as a distinct species. While it is still difficult to assign one haplotype to *H. stepanowi* sensu stricto, haploclades 1 and 3 characterizing haemogregarines sampled from *M. leprosa* of Algeria and Tunisia, respectively, were previously identified in Attia El Hili et al. [16]. Conversely, haploclades 4 and 5 characterizing haemogregarines sampled from *M. leprosa* of Morocco and Tunisia, respectively, were identified here for the first time.

## 4. Discussion

Regarding the genetic diversity of *P. costata* leeches sampled in Morocco, Algeria, and Tunisia, we can consider that two species are well differentiated in the Maghreb. According to Kvist et al. [39], *P. costata* is phylogenetically split into seven independent lineages in the Northern Mediterranean basin, Tunisia, and Algeria—at least five of which correspond to distinct species based on COI genetic variations. Among the mitochondrial haploclades that were illustrated in [39], the C5 haploclade, which groups three different haplotypes characterizing leeches collected in Algeria and Tunisia, was also recovered in our study from leeches sampled in the same area (Haplotypes C5a–c), C5a–b being found in Algeria and Tunisia, and C5c being restricted to Algeria. In addition, we identified a new distinct haploclade that for convenience we named Haploclade C8. The latter also contained three distinct haplotypes (C8a–c) that characterized leeches that were collected in four distant localities of Morocco: C8a being found in three distinct localities, and C8b–c being restricted in the fourth locality. While the levels of COI variation clearly indicated that the C8 haploclade could also be considered as a distinct species, the phylogenetic relationships between all haploclades did not allow the resolution of the deepest nodes in the tree as already illustrated in [39], thus suggesting a rapid diversification of leeches. Based on phylogeographic studies of several amphibian and reptile species across their geographical distribution area in the Maghreb (Morocco, Algeria, and Tunisia), an Eastern–Western speciation pattern was evidenced for most of the investigated species [50]. This speciation pattern was very similar to the phylogeography of *M. leprosa* that supported its northwestern origin before the species colonized Europe [51]. It was impossible to consider a sister group relationship between the C5 and C8 haploclades, and at least two leech species infect *M. leprosa* in its home range of Africa; thus, it is very unlikely that leeches in the Maghreb co-diverged along with their chelonian hosts. Furthermore, C5b was found in both turtle species, i.e., *M. leprosa* and *E. orbicularis*, thus strengthening the conclusions of Vecchioni et al. [40] who clearly showed that the speciation of *P. costata* leeches infecting *E. orbicularis* and *E. trinacris* in Italy was not the result of co-divergence processes.

Regarding the COI genetic diversity of *H. stepanowi* revealed during this study on the blood of freshwater turtles from Morocco, Algeria, and Tunisia, and that already evidenced in [16], we can assume that at least five haemogregarine species are well differentiated in the Maghreb. Based on our phylogenetic tree, *H. stepanowi* sensu lato was phylogenetically split into five independent lineages in the Southern Mediterranean basin: Haploclades 2, 3, and 5 being restricted to Tunisia, Haploclade 1 to Tunisia and Algeria, and Haploclade 4 to Morocco. From the same mitochondrial marker, three independent lineages within the host *P. vogli* of South America were also reported [17]. Surprisingly, a fourth molecular lineage infecting the Rio Magdalena River turtle *Podocnemis lewyana* Duméril was found nested within that complex of haploclades [17], thus suggesting the occurrence of at least four undescribed species infecting two distinct host species (see Figure 3). While levels of COI genetic variations clearly indicate the occurrence of several haemogregarine species within the same host species in sympatry, even in syntopy, which is illustrated by numerous cases of coinfections, it is still extremely difficult to link a particular molecular lineage to a specific morphological species, as also reported in [17]. While the first molecular investigations on *Haemogregarina* have shown from nuclear markers that *H. stepanowi* possesses a wide distribution though Europe, Turkey, and the Middle East to Iran with a very low host-specificity [12], the most recent advents in molecular systematics of *H. stepanowi* have shown that this species could be actually split into a great number of distinct cryptic haemogregarine species [15,19,52,53], infecting *E. orbicularis* and *M. leprosa* on the one hand and additional hosts such as *M. rivulata* and the invasive turtle species *Trachemys scripta* (Thunberg) [19] on the other. Therefore, one may wonder whether the haemogregarine distribution is correlated to that of its definitive host, namely the *Placobdella* leech.

Investigating the links between host and parasite diversities to assess evolutionary processes requires well-resolved host and parasite phylogenies. It is also of utmost importance to set up exhaustive sampling procedures as far as possible in order to reduce the biases due to missing data in the field. Regarding leech and haemogregarine phylogenies, while a single leech species was found in Morocco (Haploclade C8), a single haemogregarine species was also evidenced in the same geographical area (Haploclade 4). Neither of these two haploclades were found in Algeria and Tunisia, suggesting a close correspondence between the parasite and its definitive host in Morocco. However, given the technical difficulties encountered in amplifying the COI of the haemoparasites, we did not obtain any information about parasite genotypes infecting turtles in Oued Zat and Oued Sebou localities—while we did for leeches. Additional data would therefore be needed to make conclusions. Concerning the eastern part of the investigated area, Algeria and Tunisia, while a single leech species was found across the sampled turtles (Haploclade C5), three distinct haemogregarine species were evidenced in the same geographical area, i.e., Haploclade 1 in Lac Noir (Algeria), Haploclade 3 in Ghayada (Tunisia), and Haploclade 5 in El Garia (Tunisia). Although none of these parasite haploclades were ever evidenced in Morocco, such as the host haploclade that was restricted to Algeria and Tunisia, it is still difficult to make conclusions regarding a close correspondence between the parasite and its definitive host in the eastern part of Maghreb because of the occurrence of the three distinct parasite lineages. However, it should be noted that no leeches were recovered from El Garia (Tunisia), which prevented linking Parasite Lineage 5 to a particular leech haploclade. Similarly, and as explained above, the difficulties encountered in amplifying the COI of haemogregarines did not allow for the genotyping of parasites in the Algerian Madjen Belhriti, Brabtia, and Lac Tonga, or Tunisian El Hania localities, while the genotypes of leeches were recovered. Additional data should help to provide conclusions regarding strict host (leech) and parasite (haemogregarine) specificity.

Although an Eastern–Western speciation pattern is apparent for both leeches and haemogregarines, with a certain level of geographical host fidelity for parasites, we cannot make definitive conclusions regarding co-speciation patterns between haemoparasites and leeches. The diversity of haemogregarines is the highest in the eastern part of the Maghreb; thus, we may also expect a higher leech diversity in that area, which is currently not the case. For this reason, deeper sampling should be conducted to collect more leeches on the one hand, and in as many localities as possible on the other. Lastly, the genotyping of haemogregarines was performed directly from blood samples of turtles; thus, we recommend that future studies genotype haemogregarines directly from leech blood extracts. Indeed, while host–parasite specificity seems to be very low within turtles, we still cannot reject the hypothesis of a very strict host–parasite specificity within leeches.

## Figures and Tables

**Figure 1 microorganisms-11-01584-f001:**
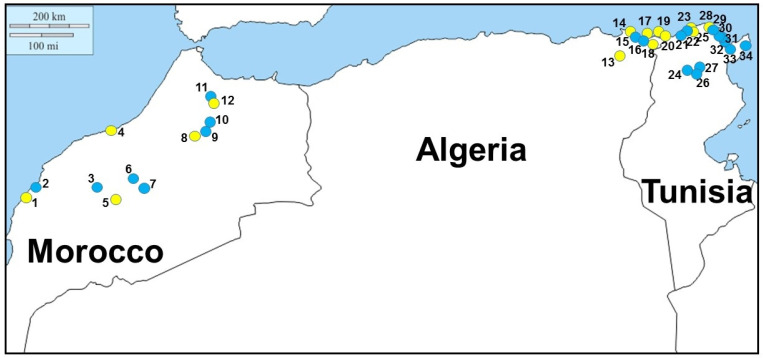
Distribution map of localities (1 to 34) where turtles were collected for haemogregarines. Yellow dots indicate field stations where leeches were found attached to limbs of turtles; blue dots indicate the absence of leeches. GPS coordinates are depicted in Table 1.

**Figure 2 microorganisms-11-01584-f002:**
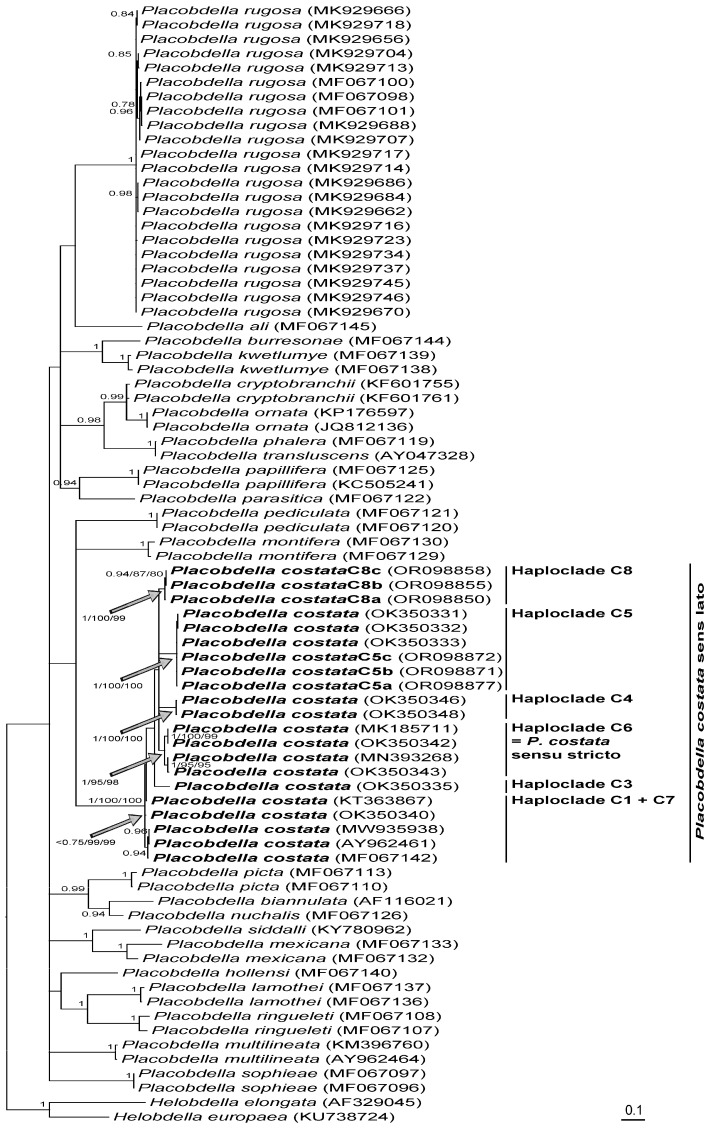
Bayesian phylogram inferred from analysis of COI, representing the phylogenetic relationships of *Placobdella costata* sensu lato. Only nodes which support posterior probabilities higher than 0.75 are shown. Values next to branches for *P. costata* (bold face) indicate, from left to right, Bayesian posterior probabilities, parsimony, and distance bootstrap proportions.

**Figure 3 microorganisms-11-01584-f003:**
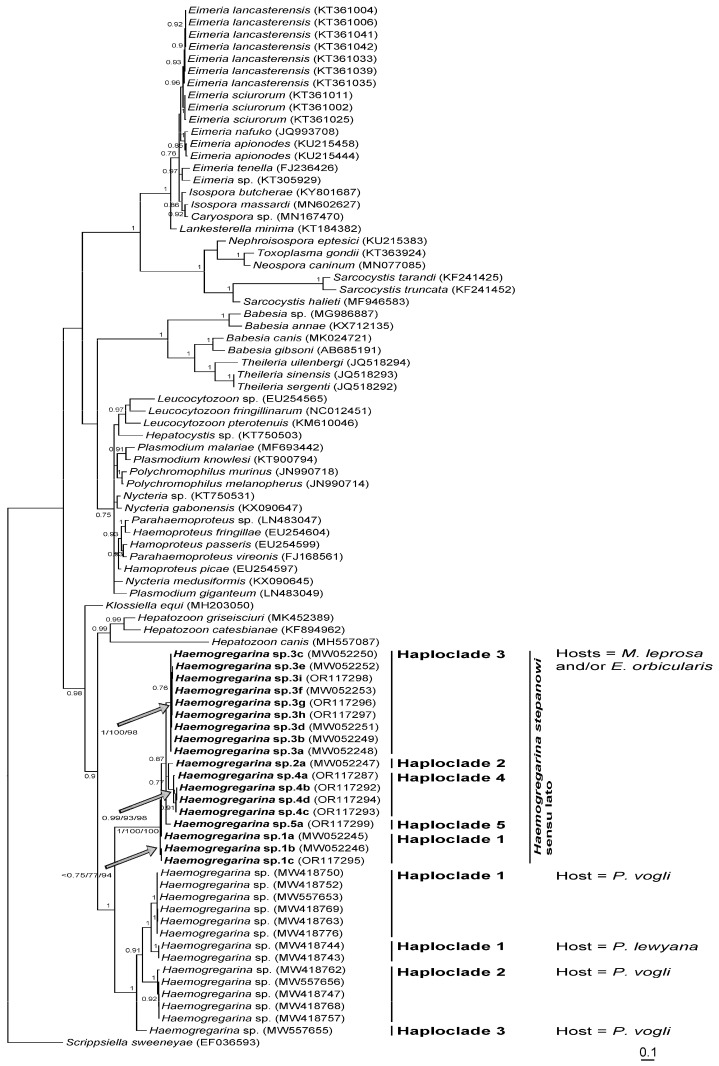
Bayesian phylogram inferred from analysis of COI, representing the phylogenetic relationships of *Haemogregarina stepanowi* sensu lato. Only nodes which support posterior probabilities higher than 0.75 are shown. Values next to branches for *H. stepanowi* (bold face) indicate, from left to right, Bayesian posterior probabilities, parsimony, and distance bootstrap proportions.

**Table 1 microorganisms-11-01584-t001:** Sampling localities of *M. leprosa* and *E. orbicularis*, leeches, and their haemogregarines.

Regional Locality	Country	GPS Coordinates	Turtle Species	Number of Turtles Examined	Presence of Leech	Number of Turtles Infected	Prevalence of Infection (%)
Oued Ksob (1)	Morocco	31°27′50.72″ N/9°45′13.94″ W	*M. leprosa*	25	+	0	0
Ain Lahjar (2)	Morocco	31°38′44.80″ N/9°35′06.69″ W	*M. leprosa*	5	−	0	0
Oued Tensift (3)	Morocco	31°44′16.08″ N/8°21′36.68″ W	*M. leprosa*	13	−	0	0
Dar Bouaza (4)	Morocco	33°32′42.03″ N/7°45′42.49″ W	*M. leprosa*	10	+	0	0
Oued Zat (5)	Morocco	31°32′34.40″ N/7°37′13.20″ W	*M. leprosa*	21	+	13	61.9
Bzou (6)	Morocco	32°04′11.60″ N/7°02′29.40″ W	*M. leprosa*	30	−	19	63.3
Oued Tissakht (7)	Morocco	32°00′39.10″ N/6°42′55.00″ W	*M. leprosa*	6	−	0	0
Oued Tigrigra (8)	Morocco	33°25′07.10″ N/5°16′29.40″ W	*M. leprosa*	20	+	16	80
Lac Zerouka (9)	Morocco	33°32′52.31″ N/5°05′44.98″ W	*M. leprosa*	20	−	0	0
Dayet Aoua (10)	Morocco	33°38′57.52″ N/5°00′46.68″ W	*M. leprosa*	26	−	0	0
Oued Boukrareb (11)	Morocco	34°04′29.92″ N/4°56′21.56″ W	*M. leprosa*	9	−	1	11.1
Oued Sebou (12)	Morocco	34°04′0.91″ N/4°54′35.98″ W	*M. leprosa*	20	+	8	40
Madjen Belhriti (13)	Algeria	36°26′09.0″ N/7°05′12.00″ E	*M. leprosa*	10	+	5	50
Lac Cité Rym (14)	Algeria	36°52′43.0″ N/7°43′36.00″ E	*E. orbicularis*	2	+	0	0
Ben Amar (15)	Algeria	36°46′59.98″ N/7°47′59.99″ E	*M. leprosa*	8	−	2	25
			*E. orbicularis*	4	−	2	50
Lac des oiseaux (16)	Algeria	36°46′44.00″ N/8°07′21.68″ E	*M. leprosa*	1	−	0	0
Lac noir (17)	Algeria	36°51′17.00″ N/8°12′20.00″ E	*M. leprosa*	12	+	5	41.6
Brabtia (18)	Algeria	36°51′09.00″ N/8°19′52.00″ E	*M. leprosa*	16	+	5	31.2
			*E. orbicularis*	4	−	3	75
Oued Bou Hchicha (19)	Algeria	36°48′47.00″ N/8°24′11.00″ E	*E. orbicularis*	1	+	0	0
Lac Tonga (20)	Algeria	36°51′37.00″ N/8°29′52.00″ E	*M. leprosa*	4	−	3	75
			*E. orbicularis*	5	+	3	60
Ouechtata (21)	Tunisia	36°56′52.04″ N/9°00′14.50″ E	*M. leprosa*	13	−	0	0
Nefza (22)	Tunisia	36°59′46.06″ N/9°04′56.56″ E	*M. leprosa*	11	−	0	0
Ghayadha (23)	Tunisia	36°59′48.36″ N/9°04′25.71″ E	*M. leprosa*	10	+	6	60
Bou Rouis (24)	Tunisia	36°09′35.40″ N/9°07′20.14″ E	*M. leprosa*	8	−	0	0
El Hania (25)	Tunisia	37°05′55.40″ N/9°13′17.19″ E	*M. leprosa*	12	+	7	58.3
Ain El Melha (26)	Tunisia	36°04′42.27″ N/9°17′47.90″ E	*M. leprosa*	12	−	0	0
Oued Silian (27)	Tunisia	36°06′40.96″ N/9°23′0.45″ E	*M. leprosa*	12	−	0	0
Béni Toun (28)	Tunisia	37°11′32.55″ N/9°43′14.54″ E	*M. leprosa*	15	+	0	0
Menzel Bourguiba (29)	Tunisia	37°08′29.29″ N/9°48′17.53″ E	*M. leprosa*	13	−	0	0
El Alia (30)	Tunisia	37°11′19.10″ N/10°02′06.75″ E	*M. leprosa*	14	−	0	0
El Garia (31)	Tunisia	37°13′18.26″ N/10°03′12.56″ E	*M. leprosa*	12	−	8	66.6
Kalaate Andalouss (32)	Tunisia	37°04′17.64″ N/10°07′11.50″ E	*M. leprosa*	11	−	0	0
Bou Achir (33)	Tunisia	36°27′28.24″ N/10°21′05.52″ E	*M. leprosa*	3	−	0	0
Oued Chiba (34)	Tunisia	36°39′10.59″ N/10°54′31.59″ E	*M. leprosa*	4	−	0	0

Note: Numbers in brackets refer to localities mapped in Figure 1.

**Table 2 microorganisms-11-01584-t002:** Genetic diversity of leeches.

Regional Locality	Leech Sample Code	Turtle Species	Turtle Sample Code	Leech 12S GenBank Accession Number	Haploclade	Haplotype
Oued Ksob (1)	MiAPL-7 *	*M. leprosa*	Ml-57	OR098850 *	C8	C8a
Oued Ksob	MiAPL-8	*M. leprosa*	Ml-74	OR098851	C8	C8a
Oued Ksob	MiAPL-27	*M. leprosa*	Ml-63	OR098852	C8	C8a
Oued Ksob	MiAPL-28	*M. leprosa*	Ml-68	OR098853	C8	C8a
Oued Zat (5)	MiAPL-10	*M. leprosa*	Ml-113	OR098854	C8	C8a
Oued Tigrigra (8)	MiAPL-1 *	*M. leprosa*	Ml-151	OR098855 *	C8	C8b
Oued Tigrigra	MiAPL-12	*M. leprosa*	Ml-157	OR098856	C8	C8b
Oued Tigrigra	MiAPL-16	*M. leprosa*	Ml-152	OR098857	C8	C8b
Oued Tigrigra	MiAPL-30 *	*M. leprosa*	Ml-163	OR098858 *	C8	C8c
Oued Sebou (12)	MiAPL-2	*M. leprosa*	Ml-175	OR098859	C8	C8a
Oued Sebou	MiAPL-13	*M. leprosa*	Ml-168	OR098860	C8	C8a
Madjen Belhriti (13)	MiAPL-39	*M. leprosa*	Ml-11	OR098861	C5	C5b
Madjen Belhriti	MiAPL-40	*M. leprosa*	Ml-4	OR098862	C5	C5b
Lac Cité Rym (14)	MiAPL-33	*E. orbicularis*	Eo-1	OR098863	C5	C5b
Lac Noir (17)	MiAPL-32	*M. leprosa*	Ml-20	OR098864	C5	C5b
Lac Noir	MiAPL-34	*M. leprosa*	Ml-22	OR098865	C5	C5b
Lac Noir	MiAPL-35	*M. leprosa*	Ml-22	OR098866	C5	C5b
Lac Noir	MiAPL-36	*M. leprosa*	Ml-22	OR098867	C5	C5b
Lac Noir	MiAPL-37	*M. leprosa*	Ml-7	OR098868	C5	C5b
Brabtia (18)	MiAPL-17	*M. leprosa*	Ml-13	OR098869	C5	C5a
Oued Bou Hchicha (19)	MiAPL-18	*E. orbicularis*	Eo-1	OR098870	C5	C5b
Lac Tonga (20)	MiAPL-15 *	*M. leprosa*	Ml-4	OR098871 *	C5	C5b
Lac Tonga	MiAPL-41 *	*E. orbicularis*	Eo-3	OR098872 *	C5	C5c
Lac Tonga	MiAPL-42	*E. orbicularis*	Eo-3	OR098873	C5	C5b
Lac Tonga	MiAPL-43	*E. orbicularis*	Eo-7	OR098874	C5	C5b
Lac Tonga	MiAPL-44	*E. orbicularis*	Eo-1	OR098875	C5	C5b
Ghayadha (23)	MiAPL-21	*M. leprosa*	Ml-102	OR098876	C5	C5a
Ghayadha	MiAPL-22 *	*M. leprosa*	Ml-108	OR098877 *	C5	C5a
El Hania (25)	MiAPL-23	*M. leprosa*	Ml-116	OR098878	C5	C5a
El Hania	MiAPL-24	*M. leprosa*	Ml-119	OR098879	C5	C5a
Béni Toun (28)	MiAPL-19	*M. leprosa*	Ml-75	OR098880	C5	C5b

Note: Numbers in brackets refer to localities mapped in Figure 1; * sequences used for phylogenetic analyses.

**Table 3 microorganisms-11-01584-t003:** Genetic diversity of haemogregarines.

Regional Locality	Haemogregarines Sample Code	Turtle Species	Turtles Sample Code	COI GenBank Accession Number	Haploclade	Haplotype
Bzou (6)	MiAJ3 *	*M. leprosa*	Ml-35	OR117287 *	4	4a
Bzou	MiAJ4	*M. leprosa*	Ml-36	OR117288	4	4a
Bzou	MiAJ5	*M. leprosa*	Ml-37	OR117289	4	4a
Bzou	MiAJ93	*M. leprosa*	Ml-24	OR117290	4	4a
Bzou	MiAJ94	*M. leprosa*	Ml-22	OR117291	4	4a
Oued Tigrigra (8)	MiAJ7 *	*M. leprosa*	Ml-155	OR117292 *	4	4b
Oued Tigrigra	MiAJ10 *	*M. leprosa*	Ml-158	OR117293 *	4	4c
Oued Tigrigra	MiAJ91 *	*M. leprosa*	Ml-148	OR117294 *	4	4d
Lac Noir (17)	MiAM17 *	*M. leprosa*	Ml-11	OR117295 *	1	1c
Ghayadha (24)	MiAK47 *	*M. leprosa*	Ml-105	OR117296 *	3	3g
Ghayadha	MiAK54 *	*M. leprosa*	Ml-102	OR117297 *	3	3h
Ghayadha	MiAK51 *	*M. leprosa*	Ml-106	OR117298 *	3	3i
El Garia (31)	MiAK-45 *	*M. leprosa*	Ml-93	OR117299 *	5	5a

Note: Numbers in brackets refer to localities mapped in Figure 1; * sequences used for phylogenetic analyses.

**Table 4 microorganisms-11-01584-t004:** Pairwise p-distances within haploclades of *Placobdella rugosa* and *Placobdella costata* sensu lato.

Haploclades	Minimum Distance	Maximum Distance	Average Distance ± SD
*P. rugosa*	0.0015	0.0182	0.009 ± 0.0042
*Placobdella costata* sensu lato			
(C1 + C7)	0.0000	0.0167	0.0109 ± 0.0055
C3	NA	NA	NA
C4	0.0030	0.0030	NA
C5	0.0015	0.0062	0.0030 ± 0.0013
C6	0.0016	0.0235	0.0144 ± 0.0083
C8	0.0000	0.0061	0.0035 ± 0.0026

Note: NA = not available.

**Table 5 microorganisms-11-01584-t005:** Pairwise p-distances between haploclades of *Placobdella costata* sensu lato.

Haploclades	Minimum Distance	Maximum Distance	Average Distance ± SD
(C1 + C7)/C3	0.0535	0.0605	0.0565 ± 0.0027
(C1 + C7)/C4	0.0607	0.0699	0.0662 ± 0.0028
(C1 + C7)/C5	0.0556	0.0738	0.0647 ± 0.0047
(C1 + C7)/C6	0.0444	0.0562	0.0517 ± 0.0030
(C1 + C7)/C8	0.0533	0.0562	0.0525 ± 0.0023
C3/C4	0.0669	0.0701	0.0685 ± 0.0016
C3/C5	0.0797	0.0828	0.0815 ± 0.0010
C3/C6	0.0556	0.0637	0.0610 ± 0.0040
C3/C8	0.0589	0.0590	0.0589 ± 0.0000
C4/C5	0.0699	0.0756	0.0724 ± 0.0018
C4/C6	0.0517	0.0581	0.0547 ± 0.0019
C4/C8	0.0532	0.0563	0.0547 ± 0.0015
C5/C6	0.0533	0.0603	0.0566 ± 0.0021
C5/C8	0.0532	0.0613	0.0571 ± 0.0023
C6/C8	0.0395	0.0471	0.0419 ± 0.0022

**Table 6 microorganisms-11-01584-t006:** Pairwise p-distances within haploclades of *Haemogregarina stepanowi* sensu lato.

Haploclades	Minimum Distance	Maximum Distance	Average Distance ± SD
1	0.0027	0.0109	0.0070 ± 0.0032
3	0.0027	0.0111	0.0043 ± 0.0029
4	0.0048	0.0288	0.0161 ± 0.0129

**Table 7 microorganisms-11-01584-t007:** Pairwise p-distances between haploclades of *Haemogregarina stepanowi* sensu lato.

Haploclades	Minimum Distance	Maximum Distance	Average Distance ± SD
1/2	0.0694	0.0700	0.0726 ± 0.0045
1/3	0.0602	0.0790	0.0693 ± 0.0046
1/4	0.0743	0.0929	0.0842 ± 0.0058
1/5	0.0705	0.0714	0.0706 ± 0.0004
2/3	0.0723	0.0876	0.0808 ± 0.0048
2/4	0.0673	0.0742	0.0702 ± 0.0024
2/5	0.0731	0.0731	NA
3/4	0.0667	0.0911	0.0821 ± 0.0060
3/5	0.0632	0.0705	0.0671 ± 0.0025
4/5	0.0679	0.0759	0.0720 ± 0.0029

Note: NA = not available.

## Data Availability

All sequences were submitted to GenBank under Accession Numbers OR098850 to OR098880 and OR117287 to OR117299.

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
