# Peer review of "Molecular Phylogenies of Leeches and Haemoparasites Infecting Freshwater Turtles in Aquatic Ecosystems of Northern Africa Suggest Phylogenetic Congruence between Placobdella costata Sensu Lato and Haemogregarina stepanowi Sensu Lato"

_microorganisms, 2023, doi:10.3390/microorganisms11061584_

Round 1

Reviewer 1 Report

Presented manuscript by Verneau et al. touches very interesting topic of coevolution in complex host-parasite systems. Generally, I like it and have no substantial reservation. Study somehow suffer of smaller sampling, necessary for real complex coevolutionary studies. Nevertheless, authors avoid taxonomic Big Bangs in leaches and haemogregarines, what I highly appreciate. There is no reason quickly describe new taxa, just when we discover that newly applied marker offers some variability. Personally, I would strongly suggest to keep the conservative approach, particularly in alpha taxonomy, not providing another names, which must be later synonymized.

After through application of mitochondrial marker in haemogregarines, we will see, what difference and tree structuring deserves nomenclatural outcome and which is just representing intraspecific diversity. New names right now, with current knowledge, would provide no advantage.

Further I provide some comments and notes.

L21 – Haemogregarines are

L 50 – either include lizards, or simplify all to reptiles

L 58 – Smith

L 82-83 – three times used word species, please rephrase it

L 89 – use genetic diversity instead of species diversity, or simply just “diversity”

L 95 – against??

L 106 – amphiumas = amphibians??

Page 8 down – Table heading is not easy to understand, please give attention to Table organization and placement

Captions to Table 2 and 3 – “used in genetics” sounds like used in chemistry, please rephrase this formulations, e.g. included in sequence analyses or other way as you prefer.

L 334 – costata

L 357 – other

L 362-363 – from blood turtles,….. better from blood of freshwater turtles from Morocco,…..

L 364 – regarding suggesting five new species, again, taxonomy must be more conservative to provide stable environment, so I am suggesting cautiousness, at least now, till the mitochondrial markers will provide more data and information.

As I wrote, the paper is fine and I suggest its acceptance after considering my few comments.

Author Response

Point 1. L21 – Haemogregarines are

Response 1: We do not really understand what the reviewer is expecting by “Haemogregarines are”

Point 2. L 50 – either include lizards, or simplify all to reptiles

Response 2: Unchanged as birds are not reptiles

Point 3. L 58 – Smith.

Response 3: Corrected in the text

Point 4. L 82-83 – three times used word species, please rephrase it

Response 4: Corrected in the text

Point 5. L 89 – use genetic diversity instead of species diversity, or simply just “diversity”

Response 5: Corrected in the text

Point 6. L 95 – against??

Response 6: Corrected in the text

Point 7. L 106 – amphiumas = amphibians??

Response 7: Amphiumas is correct. Sentence has been however modified

Point 8. Page 8 down – Table heading is not easy to understand, please give attention to Table organization and placement

Response 8: Corrected in the table

Point 9. Captions to Table 2 and 3 – “used in genetics” sounds like used in chemistry, please rephrase this formulations, e.g. included in sequence analyses or other way as you prefer.

Response 9: Corrected in tables 2 and 3

Point 10. L 334 – costata

Response 10: Corrected in the text

Point 11. L 357 – other

Response 11: Corrected in the text

Point 12. L 362-363 – from blood turtles,….. better from blood of freshwater turtles from Morocco,…..

Response 12: Corrected in the text

Point 13. L 364 – regarding suggesting five new species, again, taxonomy must be more conservative to provide stable environment, so I am suggesting cautiousness, at least now, till the mitochondrial markers will provide more data and information.

Response 13: Corrected in the text

Reviewer 2 Report

Congratulations on your manuscript! It is a very well-executed piece of work. However, there is limited information provided about leeches and gregarines to explain the biodiversity of aquatic vertebrates. I believe that this study is incredibly valuable and should be published, as it fills a crucial gap in the current knowledge.

Author Response

(The authors gave the same response as above.)
